# Early Doxorubicin Myocardial Injury: Inflammatory, Oxidative Stress, and Apoptotic Role of Galectin-3

**DOI:** 10.3390/ijms232012479

**Published:** 2022-10-18

**Authors:** Suhail Al-Salam, Karthishwaran Kandhan, Manjusha Sudhadevi, Javed Yasin, Saeed Tariq

**Affiliations:** 1Department of Pathology, College of Medicine and Health Sciences, United Arab Emirates University, Al Ain P.O. Box 15551, United Arab Emirates; 2Department of Internal Medicine, College of Medicine and Health Sciences, United Arab Emirates University, Al Ain P.O. Box 15551, United Arab Emirates; 3Department of Anatomy, College of Medicine and Health Sciences, United Arab Emirates University, Al Ain P.O. Box 15551, United Arab Emirates

**Keywords:** heart, doxorubicin, Galectin-3, inflammation, oxidative stress, apoptosis

## Abstract

Doxorubicin (DOXO) is an effective drug that is used in the treatment of a large number of cancers. Regardless of its important chemotherapeutic characteristics, its usage is restricted because of its serious side effects; the most obvious is cardiotoxicity, which can manifest acutely or years after completion of treatment, leading to left ventricular dysfunction, dilated cardiomyopathy, and heart failure. Galectin 3 (Gal-3) is a beta galactoside binding lectin that has different roles in normal and pathophysiological conditions. Gal-3 was found to be upregulated in animal models, correlating with heart failure, atherosclerosis, and myocardial infarction. Male C57B6/J and B6.Cg-Lgals3 <^tm 1 Poi^>/J Gal-3 knockout (KO) mice were used for a mouse model of acute DOXO-induced cardiotoxicity. Mice were given DOXO or vehicle (normal saline), after which the mice again had free access to food and water. Heart and plasma samples were collected 5 days after DOXO administration and were used for tissue processing, staining, electron microscopy, and enzyme-linked immunosorbent assay (ELISA). There was a significant increase in the heart concentration of Gal-3 in Gal-3 wild type DOXO-treated mice when compared with the sham control. There were significantly higher concentrations of heart cleaved caspase-3, plasma troponin I, plasma lactate dehydrogenase, and plasma creatine kinase in Gal-3 KO DOXO-treated mice than in Gal-3 wild type DOXO-treated mice. Moreover, there were significantly higher heart antioxidant proteins and lower oxidative stress in Gal-3 wild type DOXO-treated mice than in Gal-3 KO DOXO-treated mice. In conclusion, Gal-3 can affect the redox pathways and regulate cell survival and death of the myocardium following acute DOXO injury.

## 1. Introduction

Doxorubicin (DOXO), an anthracycline antibiotic, is an effective chemotherapeutic drug that is used in the treatment of a large number of malignancies [1]. DOXO has important therapeutic characteristics but its usage is affected because of the serious side effects; the most conspicuous is cardiotoxicity, which can manifest acutely, subacutely, or chronically, leading to left ventricular dysfunction, dilated cardiomyopathy, and heart failure [2]. The most important noxious effect of DOXO is cardiomyocyte injury with apoptotic and necrotic cell death [3]. The severity of heart disease correlates well with the cumulative dose of DOXO during the course of cancer therapy [4]. Cardiac myocytes require optimally functioning mitochondria to produce enough ATP needed for maintaining their contractile function. Mitochondrial injury is crucial to DOXO-induced cardiac dysfunction and cell death [5]. The major cause of DOXO-induced mitochondrial injury is the over-production of reactive oxygen (ROS) and nitrogen species (RNS), such as superoxide, hydrogen peroxide, hydroxyl radical, and peroxynitrite [6].

Both ROS and RNS lead to DNA damage, oxidation, nitrosylation of proteins, and peroxidation of lipids. DOXO binds to mitochondrial DNA and impairs the electron transport chain resulting in the production of ROS and decreased ATP [7].

Cardiolipin (CL) is an inner mitochondrial membrane-specific phospholipid that plays a critical role in maintaining mitochondrial bioenergetics and mitochondrial membrane potential [8]. CL is known to have a high affinity for the DOXO, resulting in extensive mitochondrial accumulation. The majority of DOXO, taken up by cells, accumulates in the nucleus, yet a significant amount of DOXO is also known to accumulate in the mitochondria [9]. CL is considered to play a critical role in the mitochondrial accumulation of DOXO due to the formation of strong complexes with both DOXO and DOXO-Fe^3+^ complex [10]. Therefore, DOXO-induced mitochondrial toxicity is partly, due to the formation of a strong DOXO-CL complex, resulting in the retention of DOXO in the inner mitochondrial membrane, permitting it to undergo continued (but unsuccessful) redox cycling, and leading to extensive oxidative damage to mitochondria [9,10].

The DNA damage stimulates the production of the ataxia telangiectasia mutated protein, which upregulates and activates p53 [11]. P53 upregulates expression of pro-apoptotic members Bax and Bad as well as increases expression of Bcl2/adenovirus E1B 19 kDa protein-interacting protein 3 (Bnip3), which can cause mitochondrial damage and necrotic cell death, as well as initiate mitophagy [12]. DNA damage and increased levels of ROS can lead to downregulation of the transcription factor GATA-4, which decreases the expression of the anti-apoptotic and anti-autophagy-initiation protein, Bcl-2 [13].

Galectin-3 (Gal-3) is a ~35 kDa protein. It is unique, chimera-like, and has one C-terminal carbohydrate recognition domain connected to a long N-terminal domain. It is found on the cell surface and within the extracellular matrix, as well as in the cytoplasm and nuclei of the cells [14].

Intracellular Gal-3 has been involved in cellular proliferation and anti-apoptotic mechanisms. Gal-3 affects K-Ras and Akt proteins and therefore regulates cellular differentiation, survival, and death [15]. While, extracellular Gal-3 mediates cell-cell adhesion, cell-matrix interaction and signaling, inflammation, and apoptosis [16]. Gal-3 has been shown to translocate either from the cytosol or from the nucleus to the mitochondria following exposure to apoptotic stimuli and block changes in the mitochondrial membrane potential, thereby preventing apoptosis [17]. Studies have also shown that Gal-3 can inhibit TNF-induced apoptosis through the activation of Akt [18]. 

Gal-3 has been implicated in the development of cardiovascular diseases. Gal-3 has been shown to play an important role in atherosclerosis. This was evident through the identification higher concentration of Gal-3 within foam cells and macrophages compared to the vascular smooth muscle cells of atherosclerotic lesions [19]. Knocking out the Gal-3 gene or using modified citrus pectin (MCP) results in a significant reduction in atherosclerotic plaque [20,21]. Our group has shown a protective role of Gal-3 at 24-h following acute MI and ischemia–reperfusion injury through its anti-apoptotic and anti-necrotic functions [22,23]. 

Gal-3 has also been implicated in the development of heart failure [24]. Plasma Gal-3 concentration correlates well with the severity of heart failure, therefore, the Food and Drug Administration (FDA) has included Galectin-3 (Gal-3) in the list of validated cardiovascular biomarkers [25]. Gal-3 inhibition through MCP reduces aldosterone-induced cardiac and renal fibrosis and improves cardiac and renal function [26].

Since cardiac toxicity related to DOXO involves oxidative stress and apoptosis at early events, and Gal-3 has potential anti-apoptotic and anti-oxidant activities at early events involving inflammation and oxidative stress [6,9,10,11,12,13], we considered that it is relevant to assess the possible protective effects of Gal-3 at early DOXO-induced cardiac injury and the mechanisms underlying these effects in mice.

We have shown a significant increase in myocardial Gal-3 in acutely injured myocardium at 24-h following myocardial infarction (MI) [27]. We hypothesized that high Gal-3 in the acute phase of DOXO-induced myocardial injury has a protective role on the heart through its anti-apoptotic and anti-necrotic functions. We have used male C57BL6 mice and Gal-3 KO mice with the same background strain to look for apoptotic and necrotic markers in the acute phase of DOXO-induced myocardial injury heart samples. This is the first study on such interactions.

## 2. Results

### 2.1. Expression of Galectin-3 in the Heart

Galectin-3 is expressed in the heart of both the B6-DOXO-treated group and the B6 sham control group. There was a significant increase in Gal-3 in the B6-DOXO-treated group when compared with the B6 sham control group, *p* < 0.0001, Figure 1.

The majority of the increase in Gal-3 is mainly related to the increase in intracellular Gal-3 because extracellular fibrosis and granulation tissue (which are the main source of extracellular Gal-3) are not formed at this time point. This has been checked and confirmed by microscopic examination of the heart sections at this time point which showed no evidence of increased fibrosis or granulation tissue. Intracellular Gal-3 has a protective effect, which is our aim in this study (to show this effect).

### 2.2. Galectin-3 Is Pro-Inflammatory in Acute Doxorubicin Cardiotoxicity

#### 2.2.1. Heart IL1-B

IL1-B expression was significantly increased in the Gal-3 wild type DOXO-treated group when compared with the Gal-3 wild type sham control group, *p* < 0.001. IL1-B was higher in Gal-3 KO DOXO-treated group than Gal-3 KO sham control group but the difference was not significant (*p* = 0.2). There was a significantly higher level of IL1-B in the Gal-3 wild type DOXO-treated group than Gal-3 KO DOXO-treated group, *p* < 0.001, Figure 2A. IL1-B is an inflammatory cytokine so its increase in the hearts of DOXO-treated mice indicates that DOXO has induced an inflammatory response. IL1-B levels were higher in DOXO-treated Gal-3 wild type mice than in DOXO-treated KO mice, indicating that Gal-3 has a pro-inflammatory role.

#### 2.2.2. Heart C-Reactive Protein

C-reactive protein (CRP) expression was significantly increased in the Gal-3 wild type DOXO-treated group when compared with the Gal-3 wild type sham control group, *p* < 0.001. CRP expression was significantly higher in Gal-3 KO DOXO-treated group than Gal-3 KO sham control group *p* < 0.01. There was a significantly higher level of CRP in the Gal-3 wild type DOXO-treated group than Gal-3 KO DOXO-treated group, *p* < 0.001, Figure 2B. CRP is an inflammatory acute phase protein so its increase in the hearts of DOXO-treated mice indicates that DOXO has induced an inflammatory response. CRP levels were higher in DOXO-treated Gal-3 wild type mice than in DOXO-treated KO mice, indicating that Gal-3 has a pro-inflammatory role.

#### 2.2.3. Heart Myeloperoxidase

Myeloperoxidase (MPO) expression was significantly increased in the Gal-3 wild type DOXO-treated group when compared with the Gal-3 wild type sham control group, *p* < 0.001. MPO expression was significantly higher in Gal-3 KO DOXO-treated group than Gal-3 KO sham control group *p* < 0.01. There was a significantly higher level of MPO in the Gal-3 wild type DOXO-treated group than Gal-3 KO DOXO-treated group, *p* < 0.05, Figure 2C. MPO is a peroxidase enzyme released from neutrophil polymorphs during tissue injury so its increase in the hearts of DOXO-treated mice indicates that DOXO has induced an inflammatory response. MPO levels were higher in DOXO-treated Gal-3 wild type mice than in DOXO-treated KO mice, indicating that Gal-3 has a pro-inflammatory role.

#### 2.2.4. Heart Lipocalin 2

Lipocalin 2 expression was significantly increased in the Gal-3 wild type DOXO-treated group when compared with the Gal-3 wild type sham control group, *p* < 0.001. Lipocalin 2 expression was significantly higher in Gal-3 KO DOXO-treated group than Gal-3 KO sham control group *p* < 0.001. There was a significantly higher level of Lipocalin 2 in the Gal-3 wild type DOXO-treated group than Gal-3 KO DOXO-treated group, *p* < 0.01, Figure 2D. Lipocalin-2, a constituent of neutrophil polymorphs secondary granules, is secreted during tissue injury, so its increase in the hearts of DOXO-treated mice indicates that DOXO has induced an inflammatory response. Lipocalin-2 levels were higher in DOXO-treated Gal-3 wild type mice than in DOXO-treated KO mice, indicating that Gal-3 has a pro-inflammatory role.

### 2.3. Galectin-3 Is Anti-Apoptotic in Acute Doxorubicin Cardiotoxicity

#### Heart Cleaved Caspase-3

Cleaved caspase-3 expression was significantly increased in the Gal-3 wild type DOXO-treated group when compared with the Gal-3 wild type sham control group, *p* < 0.001. Cleaved caspase-3 expression was significantly higher in Gal-3 KO DOXO-treated group than Gal-3 KO sham control group *p* < 0.001. There was a significantly higher level of Cleaved caspase-3 in Gal-3 KO DOXO-treated group than in Gal-3 wild type DOXO-treated group, *p* < 0.05, Figure 3. Cleaved caspase-3 is the terminal executer of apoptotic cell death so its increase in the hearts of DOXO-treated mice indicates that DOXO has increased apoptotic activity. Cleaved caspase-3 levels were lower in DOXO-treated Gal-3 wild type mice than in DOXO-treated KO mice, indicating that Gal-3 has an antiapoptotic activity.

### 2.4. Galectin-3 Has an Antioxidant Activity in Acute Doxorubicin Cardiotoxicity

#### 2.4.1. Heart GSH

GSH expression was significantly decreased in the Gal-3 wild type DOXO-treated group when compared with the Gal-3 wild type sham control group, *p* < 0.001, Figure 4A. GSH expression was significantly decreased in Gal-3 KO DOXO-treated group when compared Gal-3 KO sham control group *p* < 0.001, Figure 4A. There was a significantly higher level of GSH in the Gal-3 wild type DOXO-treated group than Gal-3 KO DOXO-treated group, *p* < 0.05, Figure 4A. GSH is an antioxidant marker therefor its decrease in the hearts of DOXO-treated mice indicates that DOXO has increased oxidative stress. However, GSH levels were higher in DOXO-treated Gal-3 wild type mice than DOXO-treated KO mice supporting the antioxidant activity of Gal-3.

#### 2.4.2. Heart Catalase

Catalase expression was significantly decreased in the Gal-3 wild type DOXO-treated group when compared with the Gal-3 wild type sham control group, *p* < 0.001, Figure 4B. Catalase expression was significantly decreased in Gal-3 KO DOXO-treated group when compared Gal-3 KO sham control group *p* < 0.001, Figure 4B. There was a significantly higher level of Catalase in the Gal-3 wild type DOXO-treated group than Gal-3 KO DOXO-treated group, *p* < 0.01, Figure 4B. Catalase is an antioxidant marker so its decrease in the hearts of DOXO-treated mice indicates that DOXO has increased oxidative stress. However, catalase levels were higher in DOXO-treated Gal-3 wild type mice than in DOXO-treated KO mice, indicating that Gal-3 has antioxidant activity.

#### 2.4.3. Heart Malondialdehyde

Malondialdehyde (MDA) expression was significantly increased in the Gal-3 wild type DOXO-treated group when compared with the Gal-3 wild type sham control group, *p* < 0.001, Figure 4C. MDA expression was significantly higher in Gal-3 KO DOXO-treated group than Gal-3 KO control group *p* < 0.001, Figure 4C. There was a significantly higher level of MDA in Gal-3 KO DOXO-treated group than in Gal-3 wild type DOXO-treated group, *p* < 0.05, Figure 4C. MDA is an oxidative stress marker so its increase in the hearts of DOXO-treated mice indicates that DOXO has increased oxidative stress. However, MDA levels were lower in DOXO-treated Gal-3 wild type mice than in DOXO-treated KO mice, indicating that Gal-3 has antioxidant activity.

### 2.5. Galectin-3 and Cardiomyocyte Damage Markers in Acute Doxorubicin Cardiotoxicity

#### 2.5.1. Plasma Lactate Dehydrogenase

Plasma lactate dehydrogenase (LDH) was significantly increased in the Gal-3 wild type DOXO-treated group when compared with the Gal-3 wild type sham control group, *p* < 0.001, Figure 5A. Plasma LDH was significantly higher in Gal-3 KO DOXO-treated group than Gal-3 KO sham control group *p* < 0.001, Figure 5A. There was a significantly higher level of plasma LDH in Gal-3 KO DOXO-treated group than in Gal-3 wild type DOXO-treated group, *p* < 0.001, Figure 5A. LDH is an intracellular enzyme that is released to the plasma during cell injury so its increase in the plasma of DOXO-treated mice indicates that DOXO has increased cell injury. However, LDH levels were lower in DOXO-treated Gal-3 wild type mice than DOXO-treated KO mice supporting a protective activity of Gal-3.

#### 2.5.2. Plasma Creatine Kinase

Plasma creatine kinase (CK) was significantly increased in the Gal-3 wild type DOXO-treated group when compared with the Gal-3 wild type sham control group, *p* < 0.001, Figure 5B. Plasma CK was significantly higher in Gal-3 KO DOXO-treated group than Gal-3 KO sham control group *p* < 0.001, Figure 5B. There was a significantly higher level of plasma CK in Gal-3 KO DOXO-treated group than in Gal-3 wild type DOXO-treated group, *p* < 0.001, Figure 5B. CK is an intracellular enzyme that is released to the plasma during cell injury so its increase in the plasma of DOXO-treated mice indicates that DOXO has increased cell injury. However, CK levels were lower in DOXO-treated Gal-3 wild type mice than DOXO-treated KO mice supporting a protective activity of Gal-3.

#### 2.5.3. Plasma Troponin I

Plasma Troponin I was significantly increased in the Gal-3 wild type DOXO-treated group when compared with the Gal-3 wild type sham control group, *p* < 0.001, Figure 5C. Plasma Troponin I was significantly higher in Gal-3 KO DOXO-treated group than Gal-3 KO sham control group *p* < 0.001, Figure 5C. There was a significantly higher level of plasma Troponin I in Gal-3 KO DOXO-treated group than in Gal-3 wild type DOXO-treated group, *p* < 0.001, Figure 5C. Troponin I is an intracellular protein of cardiac myocytes that is released to the plasma during cardiomyocyte injury, so its increase in the plasma of DOXO-treated mice indicates that DOXO has increased cell injury. However, troponin I levels were lower in DOXO-treated Gal-3 wild type mice than DOXO-treated KO mice supporting a protective role of Gal-3.

### 2.6. Galectin-3 Reduces Mitochondrial Damage as Demonstrated by Modified Gomori Trichrome Staining

Mitochondria within cardiomyocytes were stained red by using a modified Gomori trichrome stain. There was the enlargement of mitochondria in cardiomyocytes treated with DOXO in both Gal-3 wild type and Gal-3 KO mice (Figure 6). The size of mitochondria was larger in cardiomyocytes in DOXO-treated Gal-3 KO mice (Figure 6H) than DOXO-treated Gal-3 wild type mice (Figure 6G). Dead cardiomyocytes were stained bright red with the modified Gomori Trichrome stain. There was a higher number of dead cardiomyocytes in Gal-3 KO DOXO-treated mice (Figure 6C,E) than in Gal-3 wild type DOXO-treated mice (Figure 6D,F). The dead cardiomyocytes estimation was a general impression.

### 2.7. Electron Microscopic Changes

DOXO has shown ultrastructural changes in cardiomyocytes including myofibril damage, mitochondrial damage, intercalating disc damage, sarcoplasmic damage, and increased apoptotic bodies (Figure 7). The intensity of intracellular damage is higher in Gal-3 KO cardiomyocytes than in Gal-3 wild type cardiomyocytes (Figure 7). Cardiac apoptotic bodies were higher in Gal-3 KO than in Gal-3 wild type mice (Figure 7).

### 2.8. Gal-3 and Survival Rate in DOXO-Treated Mice

There was a significantly higher survival rate in DOXO-treated Gal-3 wild type mice than DOXO-treated Gal-3 KO mice (Table 1). This supports the protective effect of Gal-3.

## 3. Discussion

DOXO is an important and effective chemotherapeutic drug that is used in the treatment of many malignant tumors. Nevertheless, numerous systemic side effects limit its wide use in cancer therapy. Cardiac toxicity is one of those adverse effects that restrain its usage in treating patients with cancer [28,29]. Studies have shown that DOXO can cause dose-dependent cardiac toxicity when used above the dose range from 400 mg/m^2^–700 mg/m^2^ [29,30]. Therefore, understanding the mechanisms of cardiomyocyte injury is very important to identify ways that minimize DOXO-induced cardiotoxicity.

We have shown significant increase in Gal-3 levels in the heart of Gal-3 wild type mice following DOXO treatment when compared with Gal-3 sham control mice, which signifies exploring its role at the acute stage of DOXO-induced cardiac injury. We chose this model of a single injection and a single time point to study the role of Gal-3 at an early stage of the myocardial injury since at an early stage of myocardial injury, granulation tissue and fibrosis, which are the major source of extracellular Gal-3 secondary to cardiomyocyte injury, are not formed yet. Hence the majority of the increase in Gal-3 that has been identified is mainly related to the increase in intracellular Gal-3 because extracellular fibrosis and granulation tissue have not been formed at this time point. This has been checked and confirmed by microscopic examination of the heart sections which showed no increase in fibrosis or granulation tissue at this time point. Intracellular Gal-3 has a protective effect, which is our aim in this study (to show this effect). The protective effect of Gal-3 was clear at this timepoint as the survival rate at the end of experiments was 84% in DOXO-Gal-3 wild type mice compared to 58% in DOXO-Gal-3 KO mice (Table 1).

It has been shown that apoptosis-mediated loss of cardiomyocytes and oxidative stress are the main causes of DOXO-induced cardiomyopathy [31].

We have demonstrated a significant increase in cardiac cleaved caspase-3 in DOXO-treated Gal-3 wild type and Gal-3 KO mice. Remarkably, cleaved caspase-3 is significantly higher in the hearts of Gal-3-KO than in Gal-3 wild type DOXO-treated mice, suggesting that the loss of Gal-3 is associated with increased apoptotic cell death.

Studies have shown depletion of sarcoplasmic reticulum Ca^2+^ is one of the early events in DOXO-induced cardiomyopathy [29]. Cardiomyocyte apoptosis and heart failure are well associated with altered levels of Ca^2+^ and calmodulin-dependent kinase II [31]. Furthermore, inhibition of mitochondrial biogenesis is thought to be the mechanism behind DOXO-mediated cardiotoxicity, mainly by activating cell death pathways through inhibiting topoisomerase 2β [32].

Interestingly, troponin I, LDH, and CK, which are markers of cardiomyocyte injury, are also significantly higher in Gal-3-KO than in Gal-3 wild type DOXO -treated mice suggesting that the loss of Gal-3 is associated with increased cardiomyocyte injury.

These results support the view that the raised Gal-3 following DOXO cardiac injury may in some way play a protective role in response to this injury. Literature has many reports regarding the anti-apoptotic role of Gal-3 [22,33]. Intracellular Gal-3 has been shown to translocate either from the cytosol or from the nucleus to the mitochondria following exposure to apoptotic stimuli and block changes in the mitochondrial membrane potential, thereby preventing apoptosis [34]. Gal-3 has also been shown to translocate to the perinuclear membranes and inhibit cytochrome c release from the mitochondria and thereby inhibiting apoptosis [35].

Gomori stain has shown significant damage to the mitochondria of cardiomyocytes in DOXO-treated mice when compared to their sham control. This can be recognized through the significant increase in the size of mitochondria when compared with mitochondria in sham control mice. We also recognize larger mitochondria in cardiomyocytes in Gal-3 KO DOXO-treated mice than in Gal-3 wild type DOXO-treated mice. These results suggest loss of Gal-3 is associated with more damage to the cardiomyocytes’ mitochondria. Electron microscopy also shows more damage to cardiomyocytes and more apoptotic bodies when there is a loss of Gal-3 in DOXO-treated mice.

Gal-3 can inhibit TNF- induced apoptosis through the activation of Akt [14].

These anti-apoptotic effects of Gal-3 are further supported by the fact that overexpression of Gal-3 protects against cell damage and death by motivating mitochondrial homeostasis [17].

It has been shown that ROS plays a key role in DOXO-induced cardiotoxicity besides other mechanisms, such as mitochondrial dysfunction, a perturbation in iron regulatory protein, the release of nitric oxide, inflammatory mediators, calcium dysregulation, autophagy, and cell death are reported to play significant roles [29,36].

We have shown a significant reduction in antioxidants glutathione and catalase levels and significantly higher MDA in the hearts of both DOXO-treated Gal-3 wild type and Gal-3 KO mice when compared with their sham controls. DOXO is associated with the excessive generation of ROS with reduction of antioxidants as a result of their utilization [37].

We also have shown significantly higher glutathione and catalase levels and lower MDA levels in the hearts of DOXO-treated Gal-3 wild type mice than Gal-3 KO mice [38]. This demonstrates the increase in Gal-3 is associated with an increase in the antioxidant activity in the DOXO-injured myocardium, which is evidence supporting the protective role of Gal-3 in acute DOXO-induced myocardial injury. Glutathione and catalase are the most important cellular defense mechanisms against oxidative injury and are the major intracellular redox buffer in ubiquitous cell types [39,40]. Accumulating evidence suggests that the intracellular redox status regulates various aspects of cellular function [40]. Glutathione and catalase specifically provide significant antioxidant protection to the myocardium against DOXO injury [23,41].

Studies have shown that Gal-3 actions with regard to oxidative stress are variable; some point to its role as an inducer of ROS, but other studies explain its role as a molecule that is protective against ROS-mediated injuries. In an ischemia–reperfusion model in the kidney, it was shown that ROS production was more prominent in Gal-3 wild type mice when compared to Gal-3 knockout mice [42]. Early data have also demonstrated Gal-3 could stimulate superoxide production by neutrophils [43] and monocytes [44]. Therefore, the presence of Gal-3 produces more ROS and more antioxidant proteins.

In our study, we found higher anti-oxidant activity in the heart following DOXO injury in conjunction with less myocardial damage in Gal-3 wild type group than Gal-3 KO group. Our group has reported significantly higher levels of antioxidants in Gal-3 wild type than Gal-3 KO mice at 24-h of IR injury [23].

Vuèeviæ et al. have also presented higher antioxidant levels in Gal-3 wild type than Gal-3 KO mice [45].

There can be many explanations for this phenomenon. The function of antioxidant systems is to modify the highly reactive oxygen species to form intermediates, which no longer possess a threat to the cell. A balance is essential between oxidation and antioxidant levels in the system for healthy biological integrity to be maintained. Studies on ischemia and reperfusion have shown impaired superoxide dismutase activity and decreased cellular glutathione-to glutathione disulfide ratio, suggesting that the extent of superoxide anion radical produced at cardiac injury exceeded the capacity of endogenous cellular antioxidant systems [46]. Nevertheless, the same oxidative stress can lead to an increase in the antioxidant capacity and so the increase we see in the antioxidant enzymes may be due to the increase in the oxidative stress. This phenomenon was observed in a study by Bandeira et al. when the total SOD activity and the lipid peroxidation were higher in diabetics compared to non-diabetics [47]. Another study by Savu et al. also showed an increase in anti-oxidant capacity despite high levels of oxidative stress [48].

Regarding the role of Gal-3 in acute DOXO-induced cardiac injury, Matarrese et al. [17] have reported that Gal-3 interferes with ROS generation and so might interfere with very early stages of cell death that are associated with perturbation of mitochondrial homeostasis and subsequent formation of ROS. They also report that overexpression of Gal-3 protects cells from death through inhibition of ROS formation by promoting mitochondrial homeostasis [17]. Mukaru et al. [49] have also noted that an increase in Gal-3 is associated with increased GSH levels. Thus, the increase in the antioxidant activity linked to Gal-3, observed in the present study, may suggest two possibilities; either it is the result of a possible adaptive response, probably due to the increased production of the oxidative radicals, or due to the innate role of Gal-3 in decreasing oxidative stress.

Acute DOXO-induced cardiac injury is also associated with inflammatory response. We have shown significantly higher levels of CRP, IL-1B, MPO, and lipocalin-2 in Gal-3 wild type DOXO-treated mice than Gal-3 KO DOXO-treated mice supporting the proinflammatory role of Gal-3. Studies have shown that genetic and therapeutic inhibition of Gal-3 by using modified citrus pectin is associated with a reduction of inflammatory response, supporting our findings [50,51,52].

We have identified that knocking out the Gal-3 gene results in a significant decrease in heart inflammatory markers as well as a significant increase in cardiac cell death. This paradox is very interesting and suggests multiple roles of Gal-3 in DOXO-induced cardiomyocyte injury and that although knocking out Gal-3 is associated with lowering inflammatory response; it leads to an increase in cell death through increased apoptotic rate and decreased antioxidant activity. All these are pointing towards a protective role of Gal-3 in acute DOXO-induced cardiac injury. Although Gal-3’s role as a pro-inflammatory mediator is shown in our work and in previous studies [53,54,55], we do not have a clear mechanistic understanding as to how this role reconciles with its protective role following DOXO treatment.

Limitations of this study include the use of a single DOXO dose and a single timepoint of 5 days following DOXO injection. Due to these limitations, dose-dependent DOXO effects in Gal-3 knockout mice, or a time-dependent shift in cellular or pathway responses to DOXO treatment of Gal-3 knockout mice may not have been detected. These limitations will be addressed in future studies.

## 4. Materials and Methods

### 4.1. Mouse Model of Doxorubicin Cardiac Toxicity

We used a mouse model of acute doxorubicin cardiac toxicity, which has been extensively described in the literature [7]. C57BL/6 male mice and B6.Cg-Lgals3 <^tm 1 Poi^>/J homozygous genotype—Gal-3 knock-out (KO) male mice [56], purchased from Jackson Laboratories, weighing 25–30 g, 12–16 weeks, were used in this experiment. Mice were maintained on a standard diet and were housed five per cage under a 12-h light and dark schedule for at least 1 week before DOXO administration. DOXO (Aldrich, Milwaukee, WI, USA) was freshly prepared on the day of administration in sterile normal saline at a concentration of 0.5 mg/mL. DOXO was given as a single intraperitoneal (IP) injection of 20 mg/kg. Mice were given either DOXO or vehicle (normal saline) after which the mice had free access to food and water. The method of euthanasia was started with intraperitoneal injections of anesthetic drugs, which included a combination of Ketamine (100 mg/kg) and Xylazine (10 mg/kg), then hearts were dissected and blood samples were collected via cardiac puncture 5 days after DOXO administration. Resected hearts were washed in ice cold phosphate-buffered saline (PBS), immediately frozen in liquid nitrogen, and later stored in −80 °C freezer. Collected blood was centrifuged at 3000 RPM for 15 min. The plasma was collected, aliquoted, and stored at −80 °C until further analysis. In addition, another set of heart samples from the same groups were fixed in 10% buffered formal saline for 24 h.

#### Experimental Groups

Group A: Male C57BL/6 mice (*n* = 12), DOXO 20 mg/kg, was administered once intraperitoneally. Animals were sacrificed after 5 days of the administrative dose.

Group B: Male B6.Cg-Lgals3 <^tm 1 Poi^>/J homozygous genotype—Gal-3 knock-out (KO) mice (*n* = 12), DOXO 20 mg/kg, was administered once intraperitoneally.

Animals were sacrificed after 5 days of the administrative dose.

Group C: Male C57BL/6 mice (*n* = 12), normal saline was administered once intraperitoneally. Animals were sacrificed after 5 days of the administrative dose.

Group D: Male B6.Cg-Lgals3 <^tm 1 Poi^>/J homozygous genotype—Gal-3 knock-out (KO) mice (*n* = 12), normal saline was administered once intraperitoneally. Animals were sacrificed after 5 days of the administrative dose.

### 4.2. Tissue Processing

Hearts were excised, washed with ice-cold phosphate buffer saline (PBS), blotted with filter paper and weighed. Each heart was sectioned into coronal slices of 2 mm thickness then ‘cassetted’ and fixed directly in 10% neutral formalin for 24 h, which was followed by dehydration in increasing concentrations of ethanol, clearing with xylene and embedding with paraffin.

#### 4.2.1. Modified Gomori Trichrome Staining of Mitochondria

Sections were stained with the modified Gomori trichrome method to demonstrate the mitochondria using standard procedures. Five-um sections were deparaffinized with xylene and rehydrated with graded alcohol. Sections were immersed in Harris Hematoxylin for 5 min then wash with tap water until the water is clear. Sections were then immersed in Gomori trichrome stain for 10 min. Sections were then differentiated with few dips of 0.2% acetic acid. Sections were then immersed directly into 95 % alcohol and dehydrated in ascending alcohol solutions. Sections were then cleared with xylene and mounted with dibutyl phthalate polystyrene xylene (DPX).

#### 4.2.2. Electron Microscopic Study

Samples were immediately immersed in McDowell and Trump fixative for 3 h at 25 °C. Tissues were then rinsed with ethanol and propylene oxide, infiltrated, embedded in Agar-100 epoxy resin, and polymerized at 65 °C for 24 h. Blocks were then trimmed and semithin and ultrathin sections were cut with Reichert Ultracuts, ultramicrotome. The semithin sections (1 um) were stained with 1% aqueous toluidine blue on glass slides. The ultrathin sections (95 nm) on 200 mesh Cu grids were contrasted with uranyl acetate followed by lead citrate double stain. The grids were examined and photographed under a Philips CM10 transmission electron microscope.

### 4.3. Protein Extraction from Samples

Total protein was extracted from heart samples by homogenization with lysis buffer and collecting the supernatant after centrifugation. For total tissue homogenate, the heart samples were thawed, weighed and put in cold lysis buffer containing 50 mM Tris, 300 mM NaCl, 1 mM MgCl_2_, 3 mM EDTA, 20 mM β-glycerophosphate, 25 mM NaF, 1% Triton X-100, 10% *w*/*v* Glycerol and protease inhibitor tablet (Roche Complete protease inhibitor cocktail tablets). The hearts were homogenized on ice by a homogenizer (IKA T25 Ultra Turrax, IKA^®^-Werke GmbH & Co. KG, Germany). The samples were then centrifuged at 14,000 RPM for 15 min at 4 °C, supernatant, collected, aliquoted and stored at −80 °C until further analysis. Total protein concentration was determined by the BCA protein assay method (Thermo Scientific Pierce BCA Protein Assay Kit, Waltham, MA, USA).

### 4.4. Enzyme-Linked Immunosorbent Assay (ELISA)

Myocardial concentrations of Gal-3, C—reactive protein (CRP), IL1B, IL-6, myeloperoxidase, lipocalin-2, cleaved caspase-3, total glutathione, catalase, malondialdehyde (MDA), and plasma troponin I levels were measured by ELISA according to the manufacturer’s instructions. The levels were normalized to total protein concentrations. Briefly, 96-well plates (Nunc-Immuno Plate MaxiSorp Surface (NUNC, A/S, Roskilde, Denmark), were coated with antibodies specific for our proteins of interest. Biotinylated detection antibody and streptavidin-conjugated horseradish peroxidase will be used for the detection of captured antigens. The plates between steps were aspirated and washed 3 times using an ELISA plate washer (BioTek ELx50, Thermo Fisher Scientific, Waltham, MA, USA). Captured proteins were visualized using tetramethylbenzidine (TMB)/hydrogen peroxide. Absorbance readings were made at 450 nm, using a 96-well plate spectrophotometer (BioTek ELx800, Thermo Fisher Scientific, Waltham, MA, USA). Concentrations of the samples were determined by interpolation from a standard curve. Standards and samples were assayed in duplicate.

### 4.5. Plasma Samples Analysis

Plasma levels of lactate dehydrogenase and creatine kinase were measured using an automated analyzer Integra 400 Plus (Roche Diagnostics, Mannheim, Germany).

### 4.6. Statistical Analysis

Statistical analysis was performed by using GraphPad Prism Software version 5. Data were presented in mean ± standard error (S.E). Comparisons between groups were performed by one-way analysis of variance (ANOVA), followed by Newman–Keuls multiple range tests. The comparison between two groups was performed by Student’s *t*-test. *p* values < 0.05 will be considered significant.

## 5. Conclusions

Gal-3 can affect the redox pathways and regulate cell survival and death of the myocardium following acute DOXO injury.

## Figures and Tables

**Figure 1 ijms-23-12479-f001:**
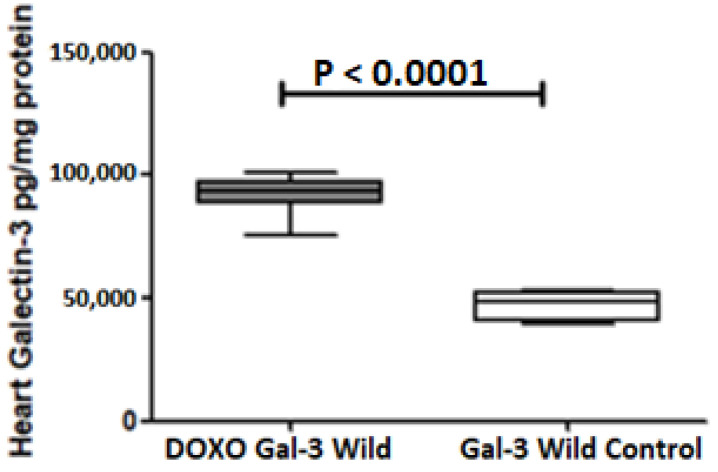
The graph represents heart concentrations of Gal-3 at 5 days following DOXO treatment in Gal-3 wild type mice compared to their control sham. *p* value < 0.05 is statistically significant.

**Figure 2 ijms-23-12479-f002:**
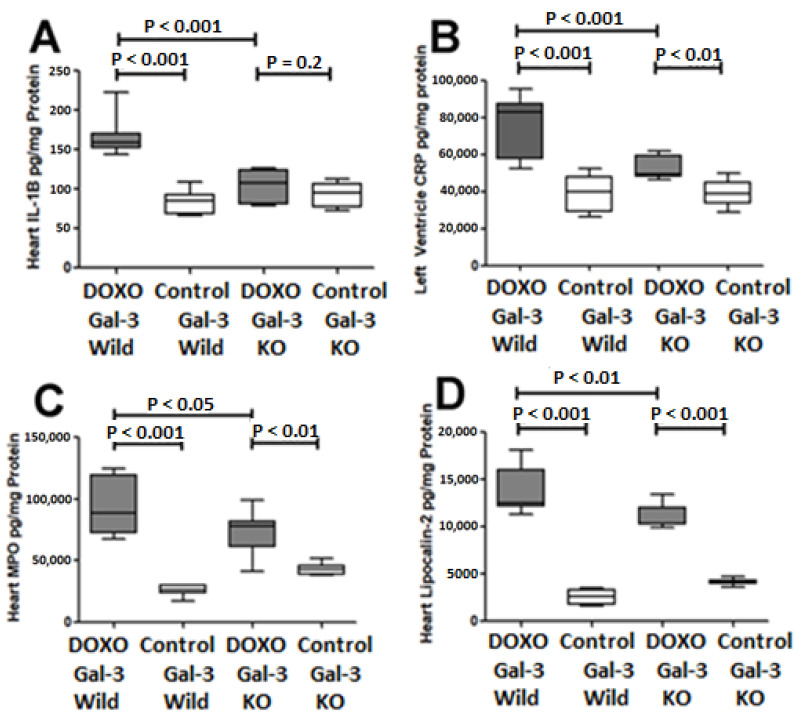
Expression of inflammation markers in DOXO-treated mice. Graphs represent cardiac concentrations of (**A**) IL-1B, (**B**) CRP, (**C**) MPO, and (**D**) lipocalin-2 at 5 days following DOXO treatment in Gal-3 wild type and Gal-3 KO mice compared to sham-treated controls. *p*-values < 0.05 were considered statistically significant.

**Figure 3 ijms-23-12479-f003:**
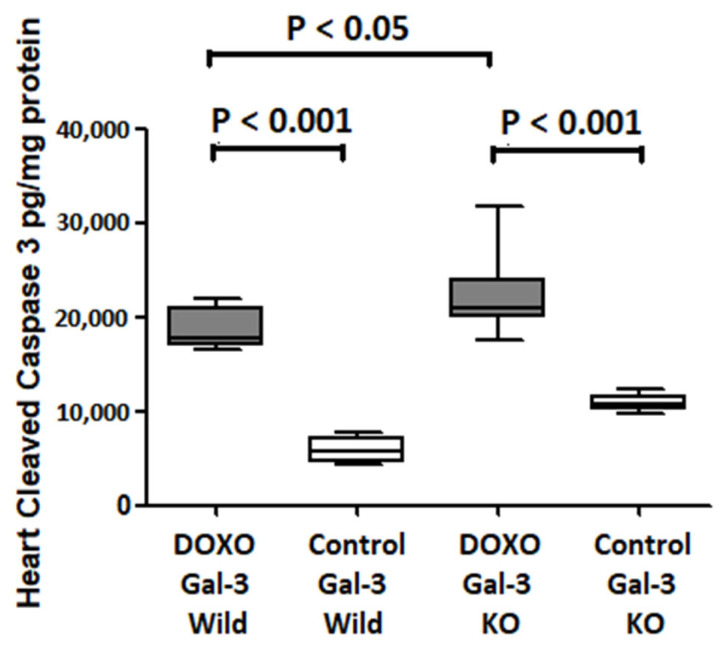
The graph represents heart concentrations of cleaved caspase-3 at 5 days follow DOXO treatment in Gal-3 wild type and Gal-3 KO mice compared to their sham controls. *p* value < 0.05 is statistically significant.

**Figure 4 ijms-23-12479-f004:**
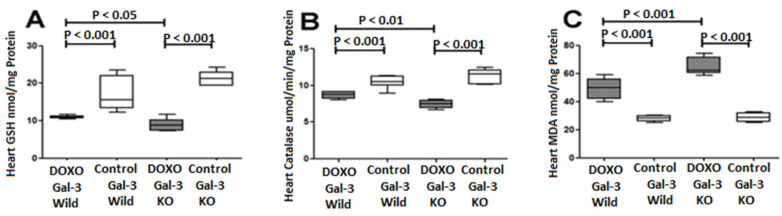
Expression of antioxidants and oxidative stress markers in DOXO-treated mice. Graphs represent cardiac concentrations of (**A**) GSH, (**B**) catalase, and (**C**) MDA at 5 days following DOXO treatment in Gal-3 wild type and Gal-3 KO mice compared to sham-treated controls. *p*-values < 0.05 were considered statistically significant.

**Figure 5 ijms-23-12479-f005:**
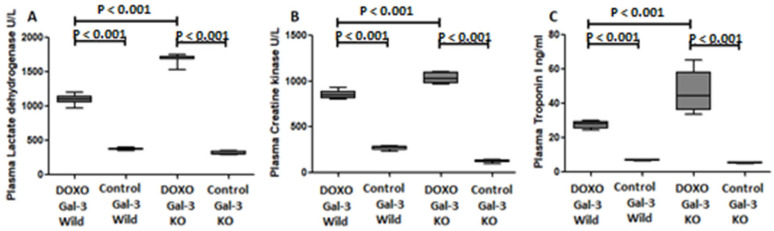
Expression of cardiomyocyte injury markers in DOXO-treated mice. Graphs represent plasma concentrations of (**A**) lactate dehydrogenase, (**B**) creatine kinase and (**C**) troponin I at 5 days following DOXO treatment in Gal-3 wild type and Gal-3 KO mice compared to sham-treated controls. *p*-values < 0.05 were considered statistically significant.

**Figure 6 ijms-23-12479-f006:**
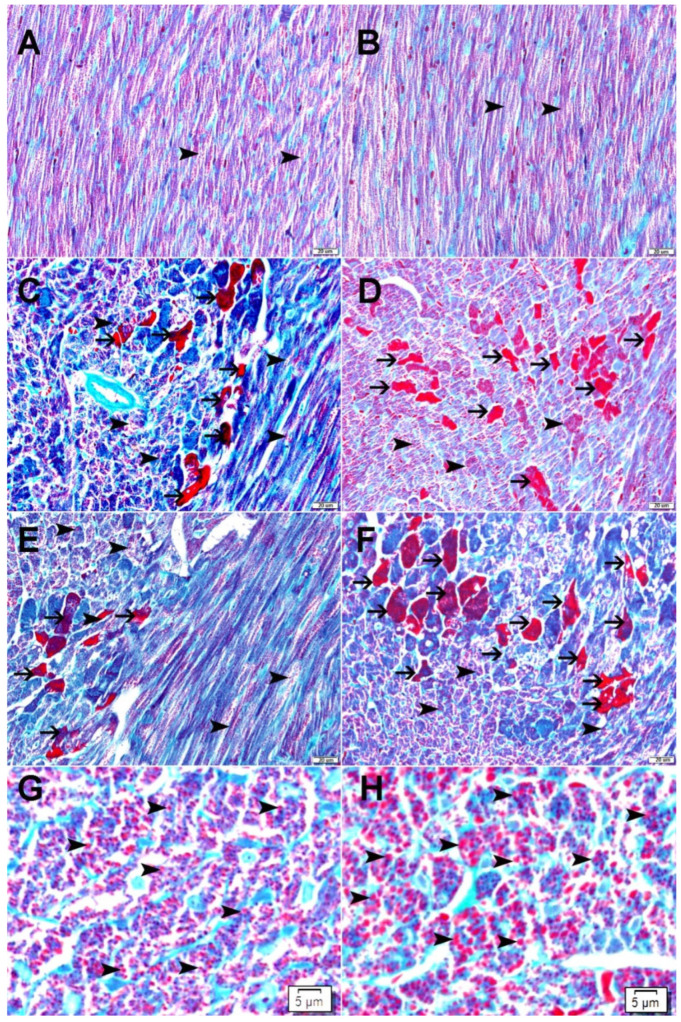
(**A**) Representative sections from the left ventricle of Gal-3 wild type saline-treated mice showing normal distribution of normal, red-stained mitochondria (arrowhead) inside cardiomyocytes. (**B**) Representative sections from the left ventricle of Gal-3 KO saline-treated mice showing normal distribution of normal, red-stained mitochondria (arrowhead) inside cardiomyocytes. (**C**) Representative sections from the left ventricle of Gal-3 wild type DOXO-treated mice showing enlarged, red-stained mitochondria (arrowhead) inside cardiomyocytes with focal severely injured cardiomyocytes (thin arrow). (**D**) Representative sections from the left ventricle of Gal-3 KO DOXO-treated mice showing enlarged, red-stained mitochondria (arrowhead) inside cardiomyocytes with a higher number of severely injured cardiomyocytes (thin arrow). (**E**) Representative sections from the left ventricle of Gal-3 wild type DOXO-treated mice showing enlarged, red-stained mitochondria (arrowhead) inside cardiomyocytes with focal severely injured cardiomyocytes (thin arrow). (**F**) Representative sections from the left ventricle of Gal-3 KO DOXO-treated mice showing enlarged, red-stained mitochondria (arrowhead) inside cardiomyocytes with a higher number of severely injured cardiomyocytes (thin arrow). (**G**) A high power magnification of representative sections from the left ventricle of Gal-3 wild type DOXO-treated mice showing enlarged, red-stained mitochondria (arrowhead) inside cardiomyocytes. (**H**) A high power magnification of representative sections from the left ventricle of Gal-3 KO DOXO-treated mice showing enlarged, red-stained mitochondria (arrowhead) inside cardiomyocytes.

**Figure 7 ijms-23-12479-f007:**
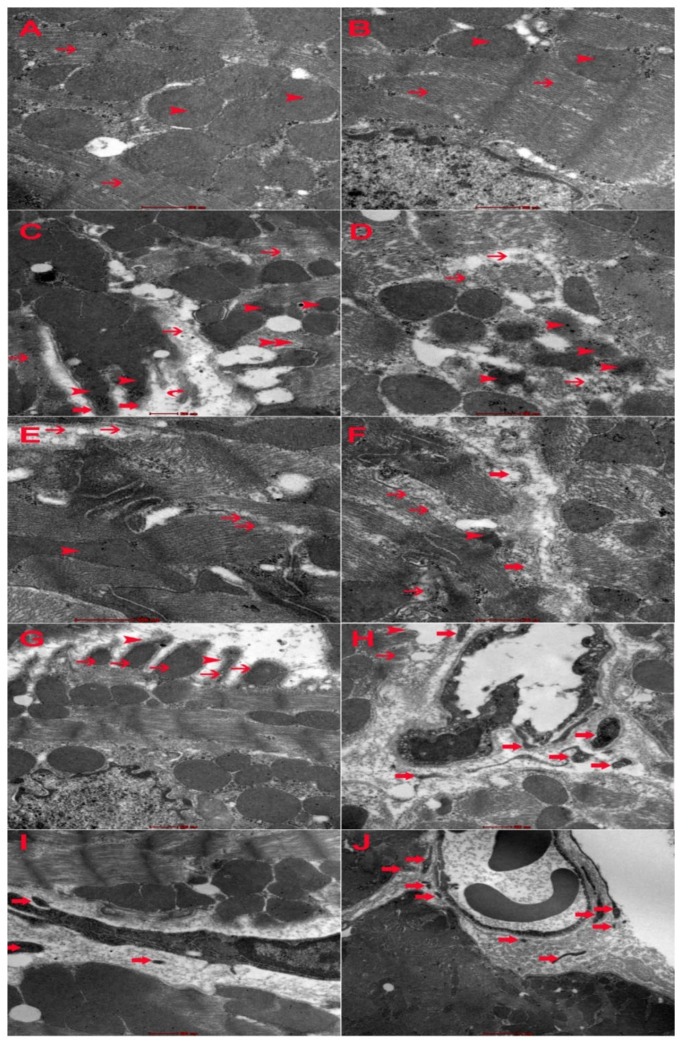
(**A**) Representative ultrathin section of cardiomyocyte from Gal-3 wild type saline-treated mice showing normal myofilaments (thin arrow) and mitochondria (arrowhead). (**B**) Representative ultrathin section of cardiomyocyte from Gal-3 KO saline-treated mice showing normal myofilaments (thin arrow) and mitochondria. (**C**) Representative ultrathin section of cardiomyocyte from Gal-3 wild type DOXO-treated mice showing damaged myofilaments (thin arrow) and damaged mitochondria (arrowhead). (**D**) Representative ultrathin section of cardiomyocyte from Gal-3 KO DOXO-treated mice showing damaged myofilaments (thin arrow) and damaged mitochondria (arrowhead). (**E**) Representative ultrathin section of cardiomyocyte from Gal-3 wild type DOXO-treated mice showing focally damaged intercalated disc (thin arrow) and damaged mitochondria (arrowhead). (**F**) Representative ultrathin section of cardiomyocyte from Gal-3 KO DOXO-treated mice showing focally damaged intercalated disc (thin arrow), damaged mitochondria (arrowhead) and damaged myofilaments (thick arrow). (**G**) Representative ultrathin section of cardiomyocyte from Gal-3 wild type DOXO-treated mice showing damaged sarcolemma (arrowhead) and damaged mitochondria (thin arrow). (**H**) Representative ultrathin section of cardiomyocyte from Gal-3 KO DOXO-treated mice showing damaged sarcolemma (arrowhead)and damaged mitochondria (thin arrow) and apoptotic bodies (thick arrow). (**I**) Representative ultrathin section of cardiomyocyte from Gal-3 wild type DOXO-treated mice showing apoptotic bodies (thick arrow). (**J**). Representative ultrathin section of cardiomyocyte from Gal-3 KO DOXO-treated mice showing apoptotic bodies (thick arrow).

**Table 1 ijms-23-12479-t001:** Survival and mortality rates in experimental groups.

Experimental Group	Number of Mice	Number of Survived Mice	%	Number Of Dead Mice	%
DOXO-treated Gal-3 wild type	12	10	84	2	16
DOXO-treated Gal-3 KO	12	7	58	5	42
Gal-3 wild type sham control	12	12	100	0	0
Gal-3 KO sham control	12	12	100	0	0

## Data Availability

The data presented in this study are available on request from the corresponding author.

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
