# Peer review of "Early Doxorubicin Myocardial Injury: Inflammatory, Oxidative Stress, and Apoptotic Role of Galectin-3"

_ijms, 2022, doi:10.3390/ijms232012479_

Round 1

Reviewer 1 Report

This is an extensive manuscript that is of interest The manuscript has a number of fundamental flaws:

1. When you refer to galectin-3, it is not clear if you refer to gal-3 in the circulation or intracellularly. Gal-3 behaves very differently and has different roles inside the cell compared to the circulation. It is more protective inside the cell and more disruptive and damaging in the circulation.

2. you do not present, cite, or discuss the multiple animal studies that show consistently that gal-KO and gal-3 inhibition (mainly with modified citrus pectin-MCP) attenuates and reverses damage to the heart, kidneys, circulation and more.

Without such discussion, the study lacks validity

Reviewer 2 Report

This manuscript presents some interesting data on the role of galectin-3 in the cardiac response to a single dose of doxorubicin, however the work overall is difficult to review in its entirety as some important details are missing and the findings could be better placed in context of the large amount of literature on the roles of galectin-3 in heart disease. The authors could consider the following comments and questions.

1. I feel that the introduction to this subject could be improved. There is a large amount of literature on the role of galectin-3 in various forms of cardiac disease, which needs to be summarised (briefly). The integral role of galectin-3 in cardiac health and disease , and how the findings of this study fit with this large amount of knowledge is not evident as either introductory (background) information or in the discussion and conclusions of the manuscript.

2. The rationale for examination of individual and groups of factors that are presented in the manuscript is not clear until the Discussion section is read. The value of the manuscript, both for researchers working in this area and for readers who are not familiar with this topic would be increased if these short descriptions explaining why the authors chose to investigate each of the factors or groups of factors was added at the start of each new results section. This would leave space in the Discussion section to place the findings of the study in context with current knowledge of doxorubicin-induced heart disease and the role of galectin-3 in heart disease and in cellular processes (e.g. inflammation) investigated in this study.

3. The authors have only used a single timepoint for their studies, which limits its broader applicability. What was the rationale for choosing 5 days (and not 1 or 3 days or other timepoints)? Would this have been the timepoint where the peak response occurred (and what evidence is there for this)? Do mice recover from this single dose of doxorubicin or do the investigators expect that the heart will not recover? These sorts of explanations (justification) and discussion points need to be added to the manuscript introduction and discussion sections. (It is relevant to discuss the limitations of the study and its findings in the Discussion section of the manuscript).

4. Details of the mice used in the study are missing. These include (i) the source of the galectin-3 knockout mice and the control mice, (ii) the age and gender of the mice (proportion of male/female mice used), (iii) how many mice were used in each of the experiments. The source of the mice (where they were purchased from) and the original publication of their derivation should both be included in the manuscript. The reference cited in the Methods section of the manuscript is incorrect for this purpose – the cited reference (reference 7) is a report of an unrelated study that used a number of genetically modified mice including a galectin-3 knockout mouse strain. Please insert a citation for the appropriate original article that describes the generation and characteristics of the galectin-3 knockout strain that was used in this study. The authors could also consider that naming the galectin-3 knockout mice “KO” in the manuscript is very confusing to scientists. KO simply stands for ‘knockout’ and any gene could be knocked out. Gal3-/- would indicate that the mice are homozygous (not heterozygous) knockout for galectin-3. (Technically, GAL-3 refers to human galectin-3, while Gal-3 refers to murine galectin-3).

5. The expression is ‘wild-type’ (not ‘wild’), hence the commonly used abbreviation ‘wt’. Please correct this error throughout the document.

6. Please re-write the figure legends. (a) Avoid repetition. For example, if p<0.05 is considered to be statistically significant, this need only be stated once at the end of the relevant figure legend(s). It does not need to be repeated for each panel in each figure as it is in Figure 2 (and other figures). If the description for each panel in a figure is identical except for the factor being measured, the sentence can be written once, listing the factor depicted in each panel (e.g. (A) factor 1, (B) factor 2, etc). A scientific English language editor will be able to assist with this. (b) Figure legends should indicate what the data represent (e.g. mean +/- SD, mean +/- SEM, etc) and how many replicates or mice were used to generate the data.

7. Avoid small repetitive sections in the manuscript text. For example, information described in sections 2.2.1 – 4 can be combined into a single section, avoiding repetitive sentences. An English language editor may be able to assist with this aspect of the manuscript.

8. Repetition of numbers in the text (e.g. “7.464 +/- 0.1909”, etc) could also be removed as the numbers don’t relate to recognisable or a “normal range” of values and they are clearly depicted in the figures. The authors could also consider how accurate their laboratory measurements are and whether 3 or 4 decimal places is appropriate (does this reflect the sensitivity and accuracy of the method/equipment used?).

9. The organisation of figure panels in figures 4 and 5 could be amended so that the blank half of the page is removed.

10. There is an error in the y-axis labelling in Figure 4C.

11. In Figure 6, panels C and F appear to be different parts of the identical image, however in the figure legend, C is supposedly from a doxorubicin-treated wild-type mouse, while F is stated to be from a galectin-3 knockout mouse. The investigators should ensure the authenticity of all figures.

12. The size of magnification bars should also be added to all images. In the text associated with Figure 6, please refer to the figure panels, not just ‘Figure 6’.

13. In this study, how were the numbers of dead cardiomyocytes calculated, or was this a ‘general impression’? This should be stated wherever quantitation of cell counts is used.

14. The electron microscopy images depicted in Figure 7 appear to be distorted (elongated). Could this please be checked?

15. Some of the articles cited in the manuscript are very old (from the 1980s) and the information derived from those studies has been updated. Accordingly, this information and citations should be updated in the manuscript text.

Reviewer 3 Report

The role of cardiolipin in DOXO-induced cardiotoxicity should be mentioned in the Introduction section.

Figure 2: The notation of the test and control group is different at the Figure 2 A and Figure caption (DOXO GAL-3 wild vs DOXO B6)

In the Results section the unit of measure of CRP, MPO, lipocalin 1, Cleaved caspase-3, GSH, malondialdehyde, etc. is missing in the text.

Row 156 and 187: Figure 4 and Figure 5 should be written instead of Figure 4.A and Figure 5. A. and in the Figure caption the A, B, C notations should be entered at the appropriate place.

The quality (pixels) of the Figures 4 A and 5 A should be increased.

Row 266 and 267: mg/m2 instead of mg/m2

Row 278 and 280: Ca2+ instead of Ca2+

Row 311-321: the authors concluded that following DOXO administration the MDA and the glutathione levels were higher compared to the control, but MDA and glutathione (reduced - GSH) levels usually have a reverse relationship, higher the MDA level leads to lower level of glutathione and vice versa (doi.org/10.3390/ijms23031255 and doi:10.3889/oamjms.2019.246), so that affirmation should be reconsidered.

The triangle interrelation between Ca2+-ATP-ROS is well known, conclusions should treat this information like it is commonly known.

Materials and method: the number of mice used in the experiment is missing (test and control group)

Row 402: abbreviation of DPX should explained

Row 425: Malondialdehyde first letter with lowercase

Row 439: the dot between the text and parenthesis should be deleted

In the whole manuscript: space between the number and unit of measure (except °C)

Round 2

Reviewer 2 Report

The authors have addressed comments made by reviewers, however, some of the changes have been inconsistently applied. I would suggest that the authors take more time to revise their manuscript and the response to reviewers’ comments. This manuscript is worthy of publication, however its presentation and the discussion of findings can be better defined so that results are not over-interpreted.

1.       1. Many of the figure legends are still repetitive and cumbersome to read. An example of a corrected legend to Figure 2 would be:- “Figure 2. Expression of inflammation markers in DOXO-treated mice. Graphs represent cardiac concentrations of (A) IL-1B, (B) CRP, (C) MPO and (D) lipocalin-2 at 5 days following DOXO treatment in Gal-3 wild-type and Gal-3 KO mice compared to sham-treated controls. P-values <0.05 were considered statistically significant.”

2.       2. Every number that is clearly depicted in figures is still repeated in the Results text, although the authors state that they have removed them. In addition, many of the numbers are to 3 or 4 decimal places and I do not believe that the assays that the authors are using are accurate to 3 or 4 decimal places.

3.       3. Because the investigators have only used a single timepoint following DOXO treatment (5 days), is it possible that the effects of DOXO treatment simply occur earlier (or later) in Gal-3 KO mice? For example, in lines 373-376, the authors state that that cleaved caspase-3 levels are higher in DOXO-treated Gal-3 KO mice, however, if Gal-3 has protective effects intracellularly, is it possible that cleavage of caspase-3 / apoptosis occurs later in Gal-3 wild-type mice? This comment is especially relevant where the investigators have used only a single marker to evaluate a cellular process (e.g. apoptosis). When discussing the results, the investigators could be mindful of the limited nature of their study design when coming to conclusions about the study and the implications of those conclusions.

Author Response

Thank you for your comments,

All comments have been addressed point by point.

Round 3

Reviewer 2 Report

The authors have addressed reviewers’ comments.

1. English language editing should be applied. There are many grammatical errors throughout the text.

2. Suggested amendment to final paragraph of the Discussion (p13).
“Limitations of this study include the use of a single DOXO dose and a single timepoint of 5 days following DOXO injection. Due to these limitations, dose-dependent DOXO effects in Gal-3 knockout mice, or a time-dependent shift in cellular or pathway responses to DOXO treatment of Gal-3 knockout mice may not have been detected. These limitations will be addressed in future studies.”

Author Response

All reviewer 2 comments have been addressed point by point.
